# Emergent Response Planning in LLMs

**Zhichen Dong**[* 1]   **Zhanhui Zhou**[* 2]   **Zhixuan Liu**[1]   **Chao Yang**[1]   **Chaochao Lu**[1]

## Abstract

In this work, we argue that large language models (LLMs), though trained to predict only the next token, exhibit emergent planning behaviors: **their hidden representations encode future outputs beyond the next token**. Through simple probing, we demonstrate that LLM prompt representations encode global attributes of their entire responses, including *structure attributes* (e.g., response length, reasoning steps), *content attributes* (e.g., character choices in storywriting, multiple-choice answers at the end of response), and *behavior attributes* (e.g., answer confidence, factual consistency). In addition to identifying response planning, we explore how it scales with model size across tasks and how it evolves during generation. The findings that LLMs plan ahead for the future in their hidden representations suggest potential applications for improving transparency and generation control.

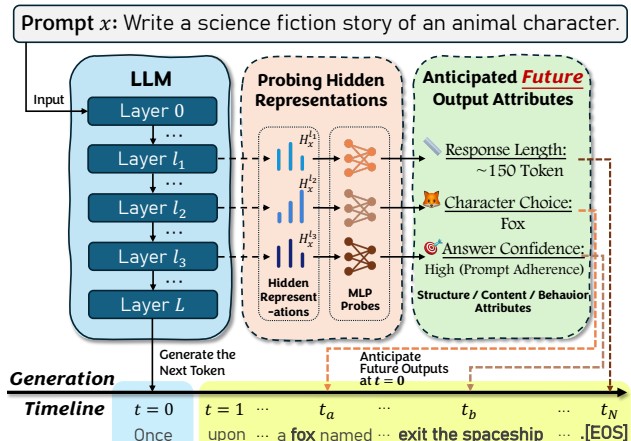

*Figure 1.* Illustration of probing LLMs for emergent response planning. After processing a prompt, hidden representations (**H**) are extracted from the LLM's layers. MLP probes analyze these representations to predict future global attributes, including structure (length), content ("Fox"), and behavior attributes (answer confidence). LLMs can anticipate their future outputs—like the "Fox" appearing at $t_a$ or the final length at $t_N$—long before generation.

## 1. Introduction

Large Language Models (LLMs) have demonstrated powerful capabilities across various tasks (Brown et al., 2020; Achiam et al., 2023; Touvron et al., 2023a; Anthropic, 2024). However, their next-token-prediction training objective leads to the view that they generate text through local, per-token prediction, without considering future outputs beyond the next immediate token (Bachmann & Nagarajan, 2024; Cornille et al., 2024) . This makes controlling the generation process challenging: *we are blind to the model's output tendency until keywords or the full response appear*. While prompt engineering and inference-time interventions (Liu et al., 2023; Li et al., 2024; Zhou et al., 2024) can guide responses, they lack insight and transparency into

---
[*]Equal contribution [1]Shanghai Artificial Intelligence Laboratory [2]Work done while at Shanghai Artificial Intelligence Laboratory. Correspondence to: Chao Yang <yangchao@pjlab.org.cn>, Zhichen Dong <dongzhichen@pjlab.org.cn>, Zhanhui Zhou <asap.zzhou@gmail.com>.

*Proceedings of the 42nd International Conference on Machine Learning*, Vancouver, Canada. PMLR 267, 2025. Copyright 2025 by the author(s).

the model's internal plan for outputs.

In this work, we argue that LLMs, though trained to predict only the next token, display emergent planning behaviors: their hidden representations encode their future outputs beyond just the next token. Specifically, we observe that LLM prompt representations encode interesting global attributes of their upcoming responses. We call this phenomenon **response planning** and classify these global attributes into three categories: *structure attributes* (e.g., response length, reasoning steps), *content attributes* (e.g., character choices in storywriting, multiple-choice answers at the end of response), and *behavior attributes* (e.g., answer confidence, factual consistency).

We empirically identify response planning by training simple probes on LLM prompt representations to predict the global attributes of their upcoming responses. We find that these probes achieve non-trivial prediction accuracy, providing strong evidence that LLMs plan at least part of their entire response in advance when they read the prompt. Through further ablations, we find that planning abilities positively scale with model size, peak at the beginning and end of responses, share certain planning patterns across

models, and exceed their self-verbalized awareness.

The contribution of our work is two-fold: (1) To our best knowledge, we introduce the first formal definition and framework of emergent response planning in LLMs. (2) We demonstrate empirically that LLMs perform emergent response planning through systematic probing experiments across various attributes types and tasks, and investigate their properties. These findings shed light on LLMs' internal mechanisms and suggest novel approaches for predicting and controlling outputs pre-generation, potentially enhancing model controllability.

## 2. Related Work

**Understanding LLM hidden representations.** LLM hidden representations encode more information than they actively use (Saunders et al., 2022; Burns et al., 2022). Patterns in these representations can be identified using linear or MLP probes (nostalgebraist, 2020; Li et al., 2022; Belrose et al., 2023; Zou et al., 2023; Ji et al., 2024) and leveraged to influence model behaviors such as truthfulness (Hernandez et al., 2023; Li et al., 2024), instruction-following (Heo et al., 2024), and sentiment (Turner et al., 2024). They are also useful for training additional regression or classification heads on transformer layers for tasks like reasoning (Han et al., 2024; Damani et al., 2024), high-dimensional regression (Tang et al., 2024), and harmful content detection (Rateike et al., 2023; MacDiarmid et al., 2024; Qian et al., 2024).

Our work also utilizes LLM hidden representations but differs in focus. Rather than using hidden states as feature extractors for external tasks, we probe model-generated data to understand how these states encode the model's own planning attributes during generation.

**Prior works exploring response planning in LLM.** Previous studies have examined whether LLMs can anticipate beyond the next token. Future Lens (Pal et al., 2023) models token distributions beyond the immediate next token using linear approximation. (Geva et al., 2023) studies how LLMs retrieve factual associations during generation, while (Men et al., 2024) extends this to Blocksworld planning, suggesting LLMs consider multiple planning steps simultaneously. (Pochinkov et al., 2024) finds that tokens at context-shifting positions may encode information about the next paragraph. (Wu et al., 2024) hypothesizes LLMs' lookahead capability and tests two mechanisms—pre-caching and breadcrumbs—in a myopic training setting.

While prior works examine relatively narrow aspects like predictions several tokens ahead or knowledge retrieval in specialized scenarios, our work delves deeper to reveal the broader response planning landscape of LLMs. We provide the first formal definition of response planning in LLMs, investigate comprehensive planning attributes, and demonstrate planning capabilities across diverse real-world tasks.

## 3. Emergent Response Planning in LLMs

If LLMs plan ahead for their entire response in prompt representations, then some global attributes of their upcoming responses can be predicted from the prompt, without generating any tokens. In this section, we first describe how existing probing techniques can investigate the global responses encoded in LLM prompt representations (Section 3.1). We then outline the setup for training our probes, including the response attributes of interest and the data collection pipeline (Section 3.2). Finally, we discuss experimental details before presenting our results (Section 3.3).

### 3.1. Probing for Future Responses

We study an $L$-layer decoder LLM $\pi(\mathbf{y} \mid \mathbf{x})$ that generates a response $\mathbf{y} = (y_1, \ldots, y_n)$ given a prompt $\mathbf{x} = (x_1, \ldots, x_m)$ sampled from a prompt distribution $p(\mathbf{x})$. During generation, the model encodes the input $(\mathbf{x} \circ \mathbf{y}_{1:t})$ into layer-wise representations $\{\mathbf{H}^l_{\mathbf{x} \circ \mathbf{y}_{1:t}}\}^L_{l-1}$, with the next token greedily decoded from the projection of final-layer representations $y_{t+1} = \arg\max(f_{\text{out}}(\mathbf{H}^L_{\mathbf{x} \circ \mathbf{y}_{1:t}}))$.

We investigate whether the prompt representations $\mathbf{H}^l_{\mathbf{x}}$, which produce the first response token $y_1$, also capture some global attributes of their upcoming response $\mathbf{y}$ (e.g., response length).

Formally, we define the *attribute rule* as $g(\mathbf{y})$, which summarizes the attributes from the generated responses (e.g., counting tokens in $\mathbf{y}$). Building on prior work on interpretability, if the prompt representations do capture these attributes, we can "probe" the hidden representations to predict the attributes without generating any response token: $h_\theta(\mathbf{H}^l_{\mathbf{x}}) \to g(\mathbf{y})$. If probing yields non-trivial predictions, we conclude that the LLM exhibits *response planning*.

### 3.2. Probing Setup

To study response planning in LLMs, we first design tasks $T = (p(\mathbf{x}), g(\mathbf{y}))$, consisting of a prompt distribution $p(\mathbf{x})$ eliciting key response attributes of interest $g(\mathbf{y})$ as probing targets. Next, we introduce the data collection pipeline for training probes.

**Task design.** The studied response attributes must be global, meaning they cannot be determined from the first response token and should ideally be distributed across the entire response. We focus on six tasks that elicit response attributes across three categories: structure, content, and behavior.

1. **Structure attributes** capture response-level features: the *response length prediction* prompts LLMs to fol-

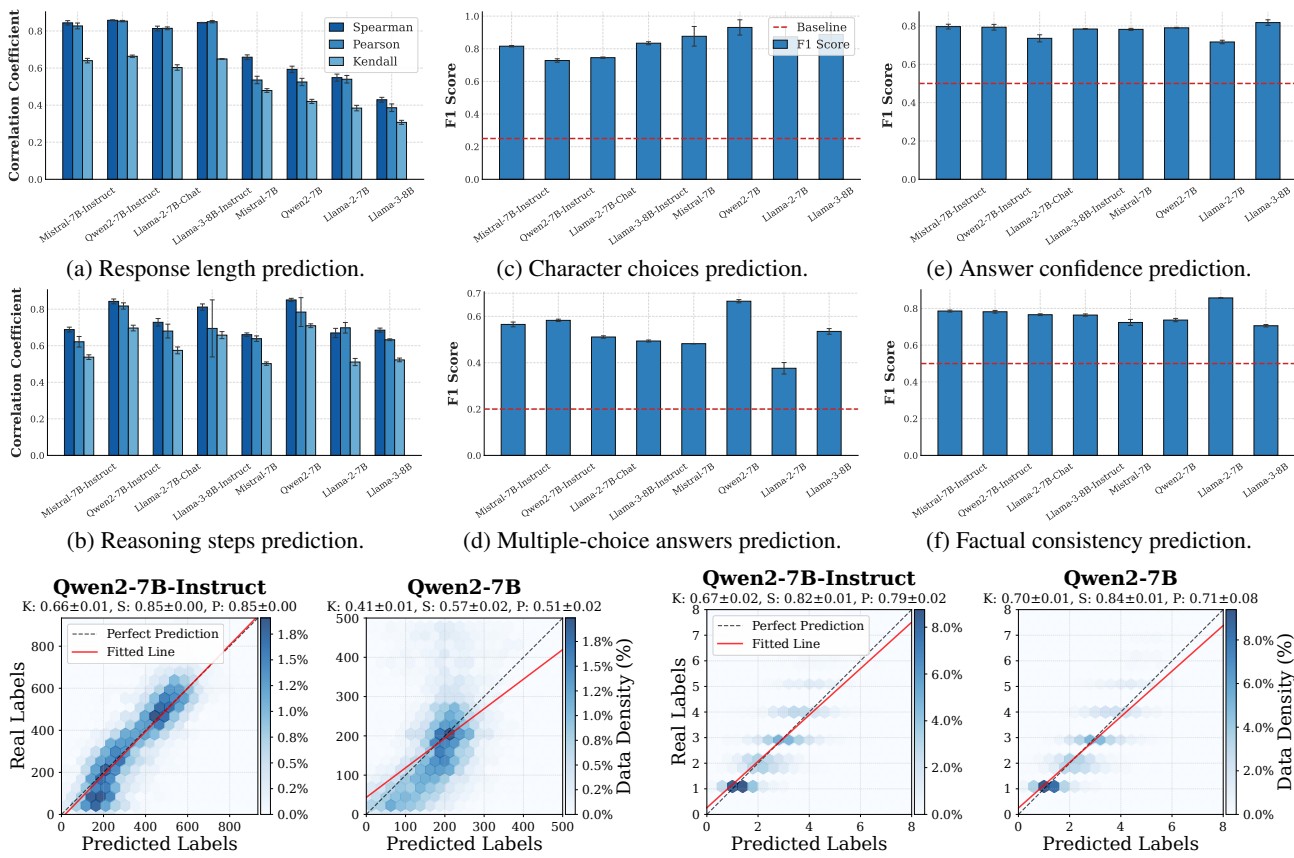

(a) Response length prediction.

(c) Character choices prediction.

(e) Answer confidence prediction.

(b) Reasoning steps prediction.

(d) Multiple-choice answers prediction.

(f) Factual consistency prediction.

(g) Example fitting results for response length prediction.

(h) Example fitting results for reasoning steps prediction.

*Figure 2.* Prediction results within the dataset. Regression tasks (response length, reasoning steps) show high accuracy and strong correlation with targets, as measured by Kendall (K), Spearman (S), and Pearson (P) coefficients. Classification tasks (character choices, multiple-choice answers, confidence, factual consistency) perform significantly above random baseline according to F1 scores. These results suggest that the model demonstrates emergent planning capabilities for future response attributes.

low human instructions, with the number of tokens counted as the probing target; the *reasoning steps prediction* prompts LLMs to solve math problems, with the number of reasoning steps as the probing target.

2. **Content attributes** track specific words appearing anywhere but not at the start of the response: *character choices prediction* prompts LLMs to write a story featuring an animal character, with the character choice as the probing target; *multiple-choice answers prediction* prompts LLMs to answer a question after reasoning (e.g., "please first explain then give your answer"), with the selected answer as the probing target.

3. **Behavior attributes** require external ground truth labels for validation: the *answer confidence prediction* prompts LLMs to answer challenging multiple-choice questions, with the correctness of answers judged by ground-truth labels as the probing target; the *factual consistency prediction* prompts LLMs to discuss and then agree/disagree with given statements, with the match between LLM's stance and statement ground-

truth validity as the probing target.

Following the prompting strategies described in each task, we carefully pair datasets with corresponding prompts. We use prompts from Ultrachat (Ding et al., 2023) and AlpacaEval (Taori et al., 2023) for response length; GSM8K (Cobbe et al., 2021) and MATH (Saxton et al., 2019) for reasoning steps; TinyStories (Eldan & Li, 2023) and ROCStories (Mostafazadeh et al., 2016) for character choices; CommonsenseQA (Talmor et al., 2019) and SocialIQA (Sap et al., 2019) for multiple-choice answers; MedMCQA (Pal et al., 2022) and ARC-Challenge (Clark et al., 2018) for answer confidence; CREAK (Onoe et al., 2021) and FEVER (Thorne et al., 2018) for factual consistency. Please see Appendix A.3.1 for more details about task design.

**Data collection.** For each task $T = (p(\mathbf{x}), g(\mathbf{y}))$, we collect datasets for probing. We sample prompts $\mathbf{x}_i$ from the prompt distribution $p(\mathbf{x})$, store prompt representations $\mathcal{H}_i = \{\mathbf{H}_{\mathbf{x}_i}^l\}_{l=1}^L$, generate responses to the prompts $\mathbf{y}_i = \arg\max \pi(\mathbf{y} \mid \mathbf{x_i})$, and store probing targets $\hat{g}_i = g(\mathbf{y}_i)$.

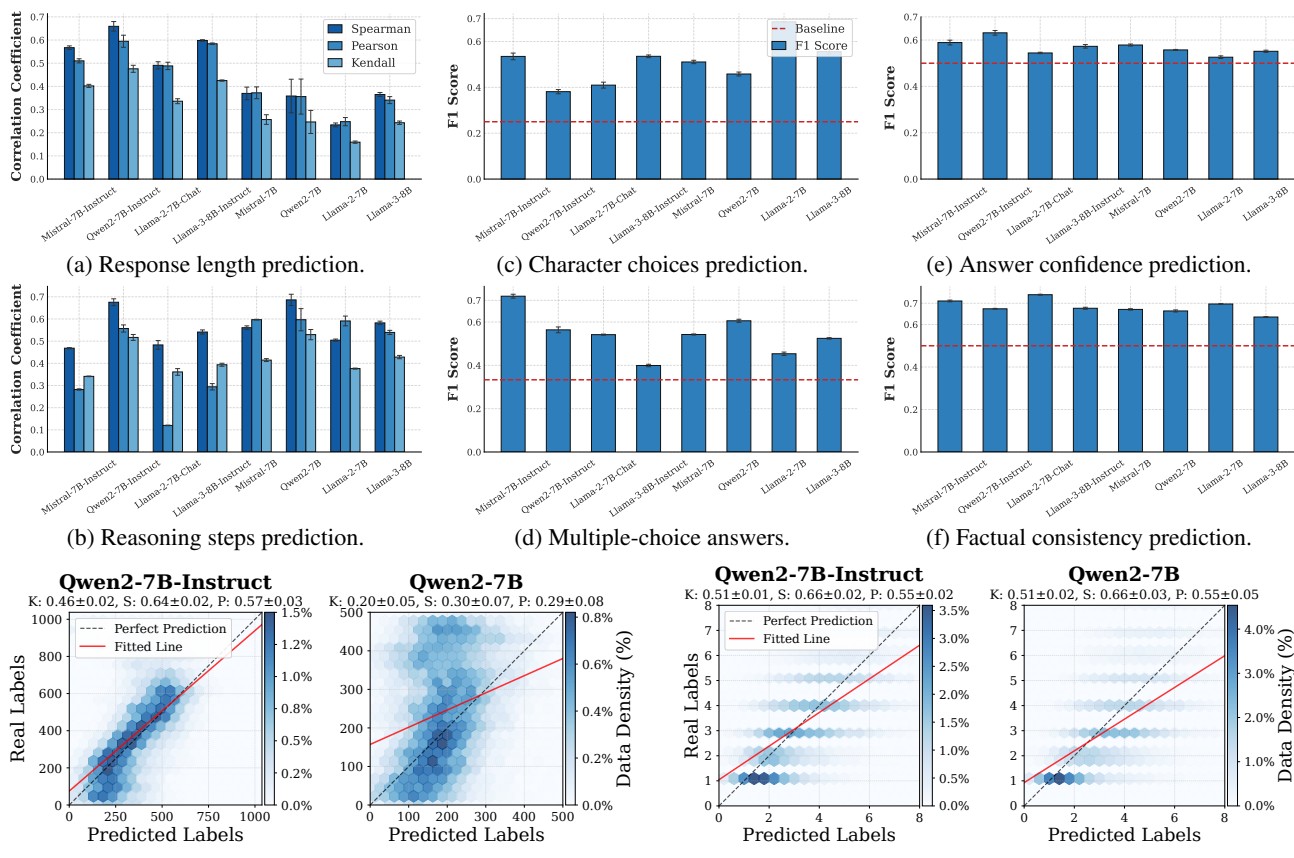

(a) Response length prediction.

(c) Character choices prediction.

(e) Answer confidence prediction.

(b) Reasoning steps prediction.

(d) Multiple-choice answers.

(f) Factual consistency prediction.

(g) Example fitting results for response length prediction.

(h) Example fitting results for reasoning steps prediction.

*Figure 3.* Cross-dataset generalization results. For regression tasks (response length, reasoning steps), correlations with targets remain strong despite reduced accuracy compared to in-dataset testing, as shown by Kendall (K), Spearman (S), and Pearson (P) coefficients. Classification tasks (character choices, multiple-choice, confidence, factual consistency) maintain above-baseline F1 scores. These results suggest the probes detect generalizable patterns rather than dataset-specific features, indicating transferable emergent planning capabilities within the task domain.

This creates a dataset of prompt representations and their future response attributes: $\mathcal{D} = \{\mathcal{H}_i, \hat{g}_i\}_{i=1}^{N}$. With this dataset, we then train a probe to predict targets from representations.

See Appendix A.3 for details on data collection, including task-specific and model-specific prompt templates, as well as data filtering and augmentation methods.

### 3.3. Experimental Details

**Probe training.** We train one-hidden-layer MLPs with ReLU activation, with hidden sizes chosen among $\mathcal{W} = \{1, 2, 4, 8, 16, 32, 64, 128, 256, 512, 1024\}$. The output size is 1 for regression and the number of classes with a softmax layer for classification. Each probe is trained for 400 epochs using MSELoss for regression and CrossEntropy-Loss for classification. Datasets are split $60:20:20$ for train-validation-test. We perform a grid search over MLP hidden sizes $\mathcal{W}$ and representation layers $\mathcal{H}$ (as inputs to the probes), reporting the test scores for the best hyperpa-

rameters. Results are averaged across three random seeds.

**Probe evaluation.** For regression tasks (response length and reasoning steps), we evaluate using Spearman, Kendall and Pearson correlation coefficients, which measure the strength and direction of monotonic (Spearman, Kendall) and linear (Pearson) relationships between predicted and target values. For classification tasks, we evaluate using F1 scores: 4-class classification for character choices, 5-class classification for multiple-choice answers, and binary classification for answer confidence and factual consistency. In our setup, accuracy aligns with F1 score for classification due to strict class balance across tasks.

**Language models.** We experiment with both instruction-tuned models (Llama-2-7B-Chat, Llama-3-8B-Instruct, Mistral-7B-Instruct, and Qwen2-7B-Instruct) and their corresponding base models (Llama-2-7B, Llama-3-8B, Mistral-7B, and Qwen2-7B). See Appendix A.1 for model details.

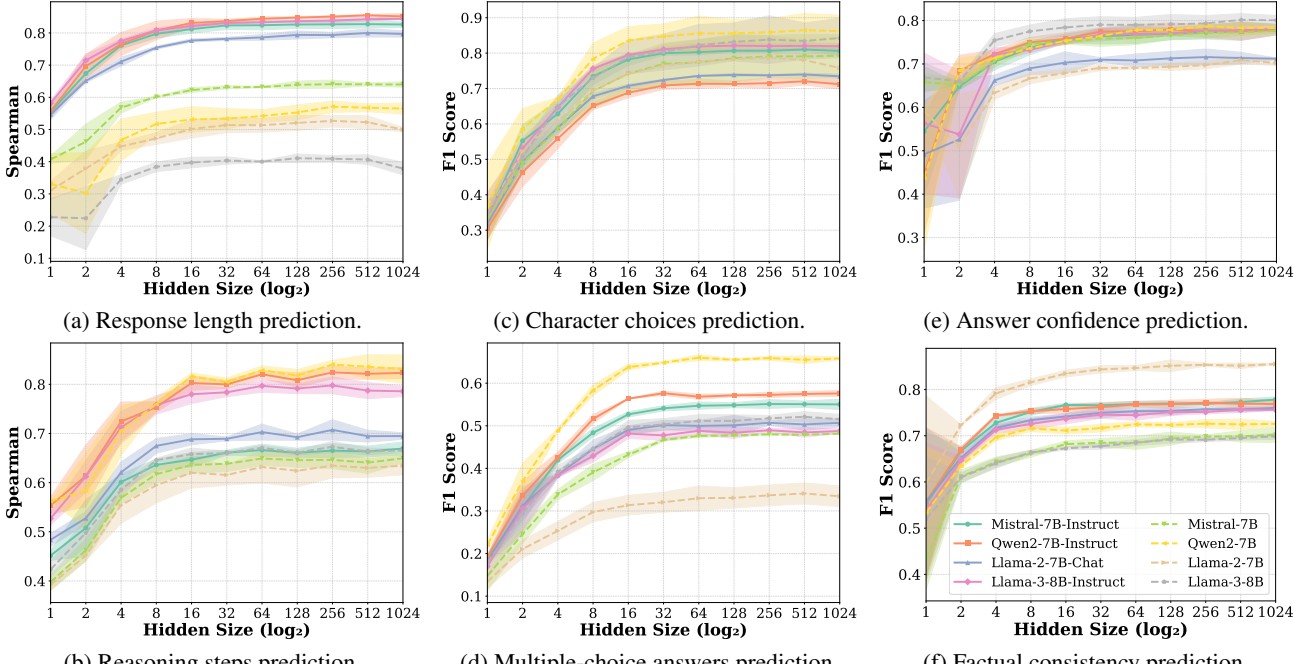

(a) Response length prediction.

(c) Character choices prediction.

(e) Answer confidence prediction.

(b) Reasoning steps prediction.

(d) Multiple-choice answers prediction.

(f) Factual consistency prediction.

*Figure 4.* Hidden-size study results. Performance of MLP probes plateaus at relatively small hidden sizes ($\leq 128$) across all tasks, with structure attributes converging around size 16, content attributes at 32, and behavior attributes at 8. This suggests a hierarchy of pattern complexity, with behavioral patterns being most accessible and content patterns requiring more sophisticated probes.

## 4. Experimental Results

In this section, we present experimental results across six tasks, showing that LLM hidden prompt representations encode rich information about upcoming responses and can be used to probe and predict global response attributes.

**Insight 1: Models present emergent planning on structure, content, and behavior attributes, which can be probed with high accuracy (Fig. 2).** Our in-dataset probing experiments (where probes are trained and tested on different splits of the same prompt dataset) reveal that both base and fine-tuned models encode structure, content, and behavior attributes, with fine-tuned models showing superior performance. For structural attributes (response length and reasoning steps; Fig. 2a, 2b), fine-tuned models exhibit strong linear correlations with ground truth, clustering around $y = x$ (with example fitting results shown in Fig. 2g, 2h), while base models show weaker but positive correlations. For content and behavior attributes (character choices, multiple-choice answers, answer confidence, and factual consistency; Fig. 2c, 2d, 2e, 2f), both model types demonstrate robust classification performance above random baselines. These findings also suggest that **models develop systematic internal planning representations for content and behavior attributes during pre-training, with structure attributes requiring additional reinforcement through fine-tuning.**

**Insight 2: The learned patterns generalize across datasets, indicating intrinsic task-related patterns rather than dataset-specific ones (Fig. 3).** Our cross-dataset experiments (training and testing probes on different prompt datasets for the same task, e.g., GSM8K→MATH or TinyStories→ROCStories) demonstrate robust generalization of learned patterns. For structure attributes (Fig. 3a, 3b), predictions maintain strong correlations with target labels despite lower accuracy compared to in-dataset testing (with example fitting results shown in Fig. 3g, 3h), with fine-tuned models showing stronger correlations than base models. Similarly, for content and behavior attributes (Fig. 3c, 3d, 3e, 3f), performance remains above baseline in cross-dataset settings. These results suggest that **probes capture generalizable task-related patterns rather than dataset-specific features, indicating that models may develop intrinsic emergent planning capabilities that transfer across different contexts within the same task domain**.

**Insight 3: Emergent planning patterns are salient across models and tasks, extractable with simple MLP probes (Fig. 4).** We investigate pattern saliency by varying the hidden size of two-layer MLP probes and measuring their average performance across model layers. Performance plateaus before hidden size 128 across all datasets, with larger sizes that can even lead to overfitting, indicating pattern saliency. The results can also indicate saliency differences across attributes: structure attributes (Fig. 4a, 4b)

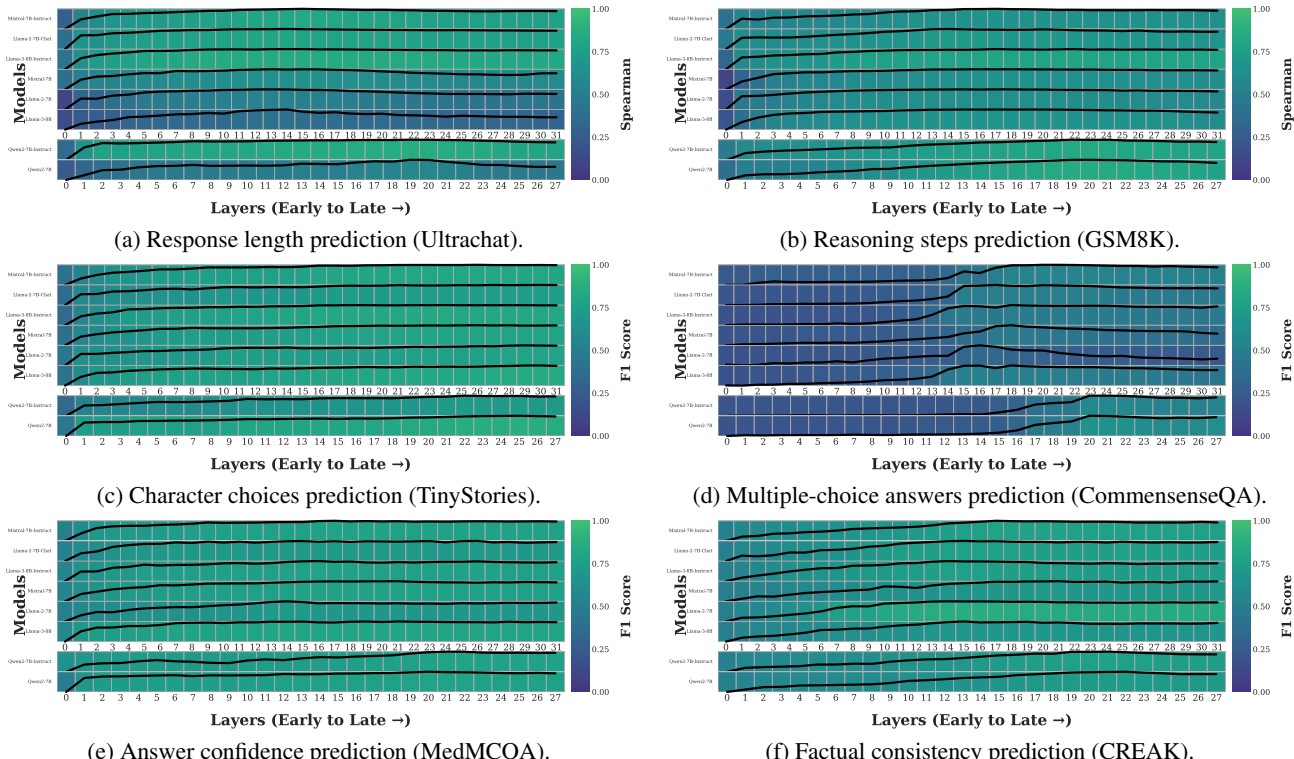

*Figure 5.* Layer-wise attribute prediction dynamics. Six subplots (one per task): Y-axis shows eight models; X-axis traces layer-wise progression (early → late). Heatmap colors indicate absolute performance (0–1, lighter = higher); black curves show row-normalized relative capability trends. Key dynamics: Structure attributes peak mid-layers, content attributes exhibit varied emergence timelines but consolidate in later layers, and behavior attributes stabilize early. Layer-wise probing reveals hierarchical organization of planning capabilities, with progressive refinement shaping final outputs.

converges around hidden size 16, content attributes (Fig. 4c, 4d) plateau around 32, and behavior attributes (Fig. 4e, 4f) plateaus at around 8, suggesting a hierarchy of representation complexity where behavioral patterns are most readily accessible, structural patterns require moderate complexity to capture, and content patterns demand the most sophisticated probe architectures. The consistent pattern across different model scales and architectures illustrates fundamental organizational principles in language model representations, suggesting that **emergent planning is an inherent property of large language models rather than an artifact of specific architectures or training procedures**.

**Insight 4: Attribute patterns accumulate and peak differently across model layers (Fig. 5).** We conduct layer-wise probing analysis (with hidden sizes optimized per layer) to understand how different attributes emerge through model layers. The results reveal distinct accumulation patterns for each attribute type. Structure attributes (Fig. 5a, 5b) show weak performance in early layers, peak in middle layers, and partially diminish in final layers, suggesting a gradual accumulation followed by refinement. Content attributes (Fig. 5c, 5d) peak in later layers, either through sudden

late-layer emergence or gradual accumulation, indicating their reliance on higher-level semantic processing. Behavior attributes (Fig. 5e, 5f) demonstrate uniform distribution across layers except for the initial few, suggesting they are fundamental properties encoded early in the model. These layer-wise patterns reveal that **(1)** different aspects of planning emerge through distinct computational paths, **(2)** the hierarchical nature of planning, from basic behavioral patterns to complex structural decisions, is reflected in the layer-wise organization, and **(3)** the emergence of these patterns through progressive transformations, rather than from initial embeddings alone, indicates that planning capabilities arise from learned computational processes rather than simple statistical correlations.

## 5. Ablation

### 5.1. Planning Ability Scales with Model Size

We analyze how emergent response planning scales across different model sizes using four model families: Llama-2-chat (7B, 13B, 70B), Llama-3-Instruct (8B, 70B), Qwen-2-Instruct (7B, 72B), and Qwen-2.5-Instruct (1.5B, 32B, 72B). Using grid search over layers and hidden sizes, we identify

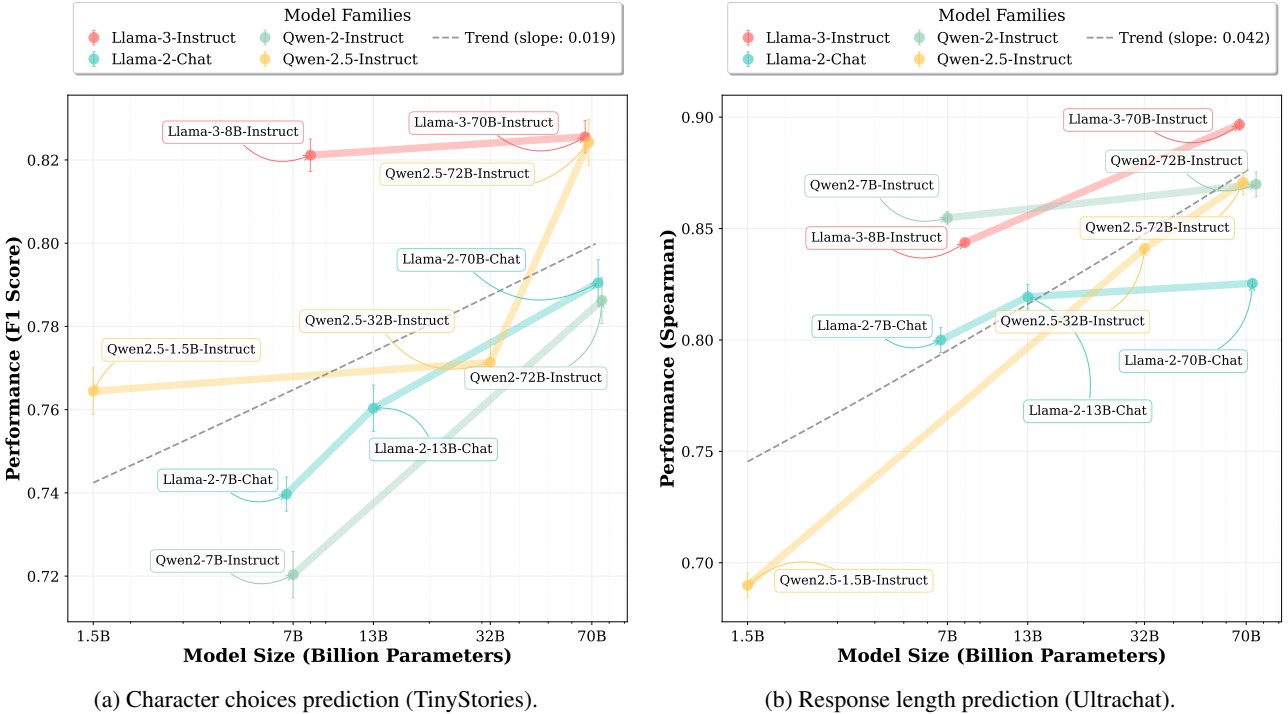

(a) Character choices prediction (TinyStories).

(b) Response length prediction (Ultrachat).

*Figure 6.* Scaling effects on planning capabilities. Evaluated across four model families (Llama-2-chat, Llama-3-Instruct, Qwen-2-Instruct, Qwen-2.5-Instruct; 1.5B–72B) using UltraChat and TinyStories, structure and content attributes show family-specific scaling: larger models within each family improve planning.

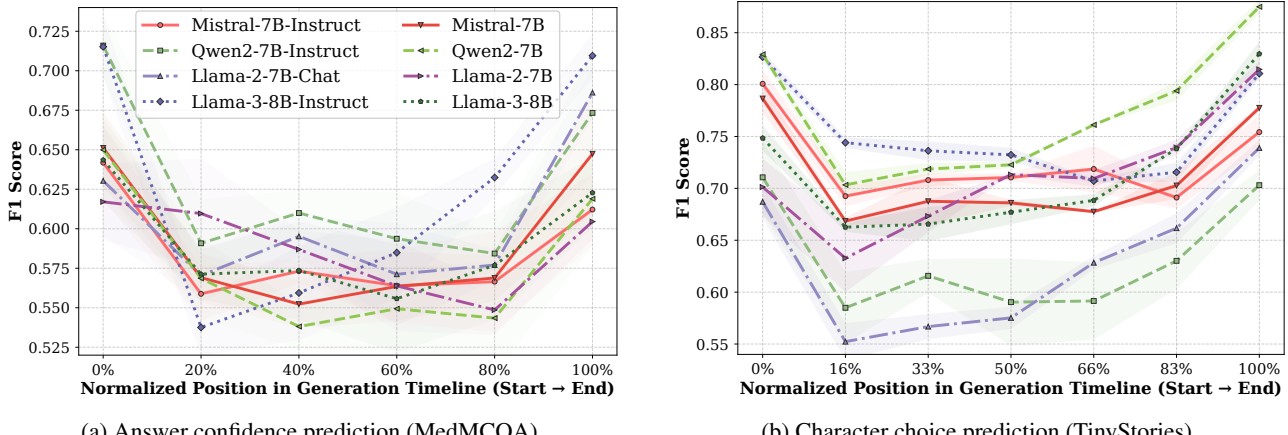

(a) Answer confidence prediction (MedMCQA).

(b) Character choice prediction (TinyStories).

*Figure 7.* Planning dynamics during generation. Probing at equidistant positions (answer confidence, character choice) shows three-phase patterns: high accuracy in early segments (global planning intent), mid-segment decline (local token focus), and late-stage recovery (contextualized refinement). This suggests models first outline global attributes, then refine locally, before finalizing coherent plans.

optimal configurations and evaluate models on UltraChat and TinyStories datasets, focusing on structure and content attributes. We exclude base models as smaller models have short context which limit few-shot prompts, while the same prompts fail to effectively prompt larger base models to follow instructions. We omit the behavior attribute type as larger models tend to give correct answers consistently, making it difficult to obtain balanced data for analysis.

Fig. 6 exhibit two key insights: **(1)** within each model fam-

ily, larger models demonstrate stronger planning capabilities, and **(2)** this scaling pattern does not generalize across different model families, suggesting that other factors like architectural differences also influence planning behavior.

## 5.2. Evolution of Planning Representations During Response Generation

We analyze how planning features evolve during generation by probing at different positions in the response sequence.

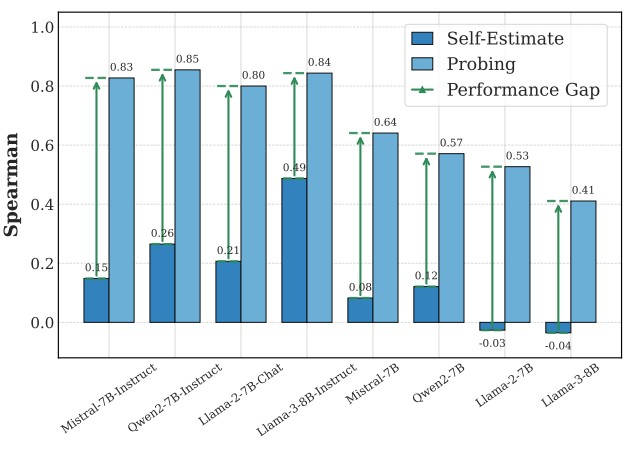

(a) Response length prediction.

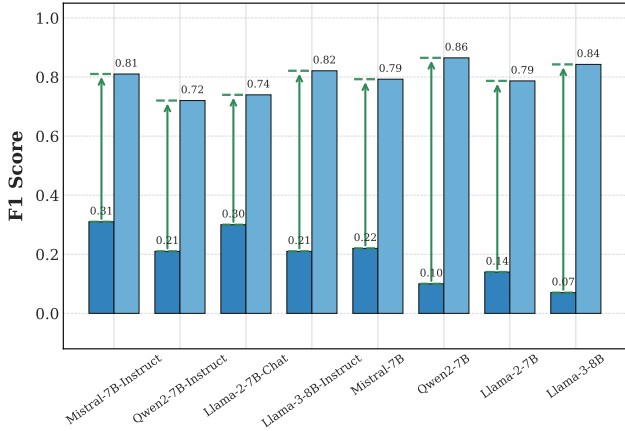

(b) Character choices prediction.

*Figure 8.* Gap between probed and verbalized results. Both tasks reveal a systematic gap between verbalized self-estimates (*Self-Estimate*) and probe-based predictions (*Probing*). Base models exhibit near or worse-than-random accuracy in self-estimation, while fine-tuned models achieve only marginal gains. This gap demonstrates that models encode richer planning information in hidden representations than they can explicitly access during generation, revealing a divide between implicit planning and explicit self-awareness.

For each response, we collect activations from the first token up to the token before attribute-revealing keywords (e.g., animal words in story character selection tasks) or throughout the entire sequence for tasks requiring external ground-truth labels (e.g., answer confidence tasks). We divide these positions into equal segments and apply probes previously trained with in-dataset settings at each division point. We conduct experiments on two datasets: TinyStories for character choice prediction and MedMCQA for answer confidence prediction. Fig. 7 reveal a distinctive pattern: **probing accuracy is high initially, decreases in the middle segments, and rises again toward the end**. This pattern suggests a three-phase planning process: **(1)** initial phase with strong planning that provides an overview of the intended response; **(2)** middle phase with weaker planning, characterized by more local, token-by-token generation; **(3)** final phase with increased planning clarity as accumulated context makes the target attributes more apparent.

### 5.3. Gap Between Probed and Verbalized Results

We investigate whether LLMs can self-estimate their response attributes when explicitly prompted, and compare these verbalized results—self-predictions obtained via direct prompting—against probe-based predictions. Experiments focus on two tasks: response length prediction (Ultrachat) and character choice (TinyStories). For verbalized predictions, we prompt LLMs in separate runs to self-estimate attributes (e.g., "Estimate your answer length in tokens using `[TOKENS]number[/TOKENS]`" for tuned models, or via few-shot examples with pre-calculated lengths for base models). Self-estimation accuracy is evaluated by comparing these outputs against actual greedy-decoded response

attributes.

Fig. 8 reveals a systematic gap: base models exhibit near or worse-than-random accuracy in self-estimation, while fine-tuned models improve only marginally, remaining far inferior to probe-based methods. This suggests **models encode more planning information in hidden representations than they can explicitly access during generation**, highlighting a divide between implicit planning and explicit self-awareness.

## 6. Discussion

### 6.1. Emergent Response Planning under Sampling

In this study, we consistently use greedy decoding to derive deterministic probing labels $\hat{g}_i = g(\mathbf{y}_i)$ for representations $\mathcal{H}_i = \{\mathbf{H}_{\mathbf{x}_i}^l\}_{l=1}^L$. But when generalizing to sampling settings, while greedy decoding simplifies sampling approximation by reflecting the LLM's most probable output, this approach may not fully capture sampling nuances. We propose two potential ways for improvement:

**Averaging**: Replace greedy labels with attribute averages over multiple sampling trials (e.g., 10 samples) to approximate expected sampling behavior.

**Distributional probing**: Train probes to predict label distributions instead of single values, capturing uncertainty inherent to sampling. While greedy decoding reflects the LLM's most probable output (approximating sampling averages), distribution-aware probing remains an open challenge, which we leave for future work.

### 6.2. Defining Planning and Addressing Spurious Correlations

A crucial consideration in defining and measuring "planning" is the potential for spurious correlations, particularly **"first-token shortcuts."** An example illustrates this: if a model is prompted in either French or English to provide a 'yes/no' answer in the same language, a probe analyzing the first token's activations might predict the final answer ("oui/non" vs. "yes/no") simply by detecting the language of the initial token. While this shows a correlation, it doesn't necessarily prove long-range planning in the sense of anticipating specific future content beyond what's implied by immediate context like language choice.

Our study addresses this by **(1)** defining planning as the encoding of long-term attributes independently of the immediate next token, and **(2)** designing prompts to actively block these shortcuts. We define planning such that the hidden representations at the first token should encode both next-token information and long-term attributes, ensuring these two information types are independent – the long-term attribute shouldn't be directly inferable from the very next token. To implement this in our experiments, we used prompt engineering. For instance, in multiple-choice tasks, we instructed models to first provide an analysis before stating their final answer. This ensures that the initial tokens (the analysis) do not inherently reveal the target attribute (the final choice), thereby helping to isolate genuine planning signals from simpler, shortcut correlations. This methodological approach is vital for ensuring that probes detect true emergent planning rather than just correlated input features.

### 6.3. Potential Applications of Emergent Response Planning in LLMs.

Our findings on LLMs' emergent response planning suggest several practical applications: **(1) Pre-generation resource allocation optimization:** Probing pre-generation representations allows proactive allocation of computational resources based on anticipated response complexity and length, enhancing inference efficiency during dynamic workloads. **(2) Early-error detection**: Early detection of behavioral attributes like low confidence could enable corrective interventions (e.g., retrieval-augmented refinement) before errors propagate. Predictive awareness of content attributes (e.g., key entities or argument trajectories) might enable real-time compliance checks with safety guidelines or domain constraints. **(3) Novel user interaction paradigms**: Predicting reasoning complexity could guide task decomposition for multi-step problems, while predicting response characteristics could improve progress indicators in interactive settings. These possibilities highlight the need for robust probing methods in deployed LLMs.

### 6.4. Future Research Directions

Several key research directions emerge from our findings: **(1) Causal mechanisms of planning:** Research could investigate whether and how planning representations directly influence token generation (e.g., via causal intervention experiments). Establishing causality is crucial for reliably leveraging these representations and understanding LLM decision-making. **(2) Leverage planning for generation control:** Future work might explore methods to detect and utilize pre-generation attribute predictions (e.g., key content points) for real-time steering. This could enable more efficient and precise control than post-hoc correction, potentially reducing computational waste and errors by allowing early adjustments based on predicted response properties. **(3) Planning in multimodal contexts:** Exploration of whether similar planning phenomena emerge in multimodal (e.g., image+text) LLMs could be valuable. Such research may provide insights into the universality of emergent planning and how modality impacts the development of these cognitive capabilities. **(4) Planning-aware training:** Developing objectives that explicitly reward alignment between early-plan encodings and final outputs (e.g., via consistency losses) represents another avenue. This may enhance coherence in complex tasks by grounding generation in initial intent.

## 7. Conclusion

In conclusion, our work reveals that LLMs have emergent response planning capabilities, with prompt representations encoding global attributes of future outputs across structure, content, and behavior attributes. These findings challenge the conventional view of LLMs as purely local predictors and provide new insights into their internal mechanisms. Though we do not focus on interpretability mechanisms to explain the causal relationship of emergent response planning, our findings open promising directions for enhancing model control and transparency, potentially enabling more effective methods for guiding and predicting model outputs before generation begins.

## Impact Statement

Our findings on LLM emergent planning raise specific considerations for model deployment. While these capabilities could enhance system reliability through better resource allocation and early warning mechanisms, they also present concerns when handling sensitive data, as these probing methods reveal aspects of the model's internal thinking or decision-making process. We encourage careful evaluation of these trade-offs when implementing probing-based monitoring systems, particularly in applications involving sensitive information.

## Acknowledgements

We would like to thank Yuyu Fan, Jiachen Ma and anonymous reviewers for their valuable feedback and discussions.

## Author Contributions

**Zhichen Dong** provided early inputs on the emergent response planning; proposed and ran the experimental tasks, and participated in writing all sections of the paper.

**Zhanhui Zhou** first pitched the research idea of emergent response planning to **Zhichen Dong**; proposed experimental tasks; made substantial writing contributions to abstract, introduction, and Section 3.

**Zhixuan Liu** provided valuable feedback throughout the project; **Chao Yang** and **Chaochao Lu** supervised and managed the group.

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

# A. Further Details on the Experimental Setup

## A.1. Model Specification

The following table lists the models and their corresponding links.

| Models | Links |
| --- | --- |
| Llama-2-7B (Touvron et al., 2023b) | https://huggingface.co/meta-llama/Llama-2-7b-hf |
| Llama-2-7B-Chat (Touvron et al., 2023b) | https://huggingface.co/meta-llama/Llama-2-7b-chat-hf |
| Llama-2-13B-Chat (Touvron et al., 2023b) | https://huggingface.co/meta-llama/Llama-2-13b-chat-hf |
| Llama-2-70B-Chat (Touvron et al., 2023b) | https://huggingface.co/meta-llama/Llama-2-70b-chat-hf |
| Llama-3-8B (AI@Meta, 2024) | https://huggingface.co/meta-llama/Meta-Llama-3-8B |
| Llama-3-8B-Instruct (AI@Meta, 2024) | https://huggingface.co/meta-llama/Llama-2-7b-hf |
| Llama-3-70B-Instruct (AI@Meta, 2024) | https://huggingface.co/meta-llama/Meta-Llama-3-70B-Instruct |
| Mistral-7B (Jiang et al., 2023) | https://huggingface.co/mistralai/Mistral-7B-v0.1 |
| Mistral-7B-Instruct (Jiang et al., 2023) | https://huggingface.co/mistralai/Mistral-7B-Instruct-v0.2 |
| Qwen2-7B (Team, 2024a) | https://huggingface.co/Qwen/Qwen2-7B |
| Qwen2-7B-Instruct (Team, 2024a) | https://huggingface.co/Qwen/Qwen2-7B-Instruct |
| Qwen2-72B-Instruct (Team, 2024a) | https://huggingface.co/Qwen/Qwen2-72B-Instruct |
| Qwen2.5-1.5B-Instruct (Team, 2024b) | https://huggingface.co/Qwen/Qwen2.5-1.5B-Instruct |
| Qwen2.5-32B-Instruct (Team, 2024b) | https://huggingface.co/Qwen/Qwen2.5-32B-Instruct |
| Qwen2.5-72B-Instruct (Team, 2024b) | https://huggingface.co/Qwen/Qwen2.5-72B-Instruct |

## A.2. Dataset Specification

The following table lists the datasets and their corresponding links.

| Datasets | Links |
| --- | --- |
| Ultrachat (Ding et al., 2023) | https://huggingface.co/datasets/stingning/ultrachat |
| AlpacaEval (Taori et al., 2023) | https://huggingface.co/datasets/tatsu-lab/alpaca |
| GSM8K (Cobbe et al., 2021) | https://huggingface.co/datasets/openai/gsm8k |
| MATH (Saxton et al., 2019) | https://huggingface.co/datasets/deepmind/math_dataset |
| TinyStories (Eldan & Li, 2023) | https://huggingface.co/datasets/roneneldan/TinyStories |
| ROCStories (Mostafazadeh et al., 2016) | https://huggingface.co/datasets/Ximing/ROCStories |

| CommonsenseQA (Talmor et al., 2019) | https://huggingface.co/datasets/tau/commonsense_qa |
| SocialIQA (Sap et al., 2019) | https://huggingface.co/datasets/allenai/social_i_qa |
| MedMCQA (Pal et al., 2022) | https://huggingface.co/datasets/openlifescienceai/medmcqa |
| ARC-Challenge (Clark et al., 2018) | https://huggingface.co/datasets/allenai/ai2_arc |
| CREAK (Onoe et al., 2021) | https://huggingface.co/datasets/amydeng2000/CREAK |
| FEVER (Thorne et al., 2018) | https://huggingface.co/datasets/fever/fever |

### A.3. Detailed Process of Response Collection and Labeling

In this section, we detail the process of collecting a dataset $\mathcal{D} = \{\mathcal{H}_i, \hat{g}_i\}_{i=1}^{N}$ for each task $T = (p(\mathbf{x}), g(\mathbf{y}))$, pairing prompt representations with their corresponding attribute labels. First, we construct the prompt distribution $p(\mathbf{x})$ to elicit responses with target attributes from the models (Sec.A.3.1). Second, we label these responses according to specific criteria $\hat{g}_i = g(\mathbf{y}_i)$ to capture their key attributes (Sec.A.3.2). Finally, we collect representations $\mathcal{H}_i = \{\mathbf{H}_{\mathbf{x}_i}^l\}_{l=1}^{L}$ for each prompt (Sec. A.3.3).

#### A.3.1. PROMPT TEMPLATES

To elicit responses with target attributes, we construct prompt distributions using carefully designed templates paired with datasets. We present the prompt templates for both chat and base models across all tasks, along with representative input-output examples.

| Task 1: Response Length |
|---|

Prompt for fine-tuned models
```
''
{data}
''
```
($\rightarrow$ Gets formatted according to model's template)

Example Response
**Data:** Why are oceans important to the global ecosystem?
**Output:** The oceans play a crucial role [...]

Prompt for base models
```
''
Q: How can cross training benefit athletes?
A: Cross training offers various benefits [...]  [END OF RESPONSE]
Q: What role does collaboration play in creativity?
A: Collaboration and originality complement each other [...]  [END
OF RESPONSE]
Q: {data}
A:
''
```

Example Response
**Data:** What are positive impacts of Reality TV?
**Output:** Reality TV provides entertainment and [...] [END OF RESPONSE]

| Task 2: Reasoning Steps |
|---|

Prompt for fine-tuned models
```
''
Provide step-by-step solution, starting with 'Step 1:'.
Problem:
{data}
```

''

($\rightarrow$ Gets formatted according to model's template)

```
Example Response
```
**Data:** Randy has 60 mango trees on his farm. He also has 5 less than half as many coconut trees as mango trees. How many trees does Randy have in all?

**Output:** Step 1: Write down the information [...]

```
Prompt for base models
''
Solve this problem step-by-step, starting with 'Step 1:'.
Few-shot examples:
Problem:  Let f(x)={ax+3 if x>2; x-5 if -2≤x≤2; 2x-b if x<-2}.
Find a+b if f is continuous.
Step 1:  At x=2:  a(2)+3=2-5 [...]  [END OF RESPONSE]
Problem:  If x=2 and y=5, find (x^4+2y^2)/6.
Step 1:  Substitute:  (2^4+2(5^2))/6 [...]  [END OF RESPONSE]
Problem:  {data}

''
```

```
Example Response
```
**Data:** Weng earns \$12 an hour for babysitting. Yesterday, she just did 50 minutes of babysitting. How much did she earn?

**Output:** Step 1: Substitute: 12(50/60) [...] [END OF RESPONSE]

```
                         Task 3:  Character Choices
```

```
Prompt for fine-tuned models
''
Here's the first sentence of a story: {data}
Continue this story with one sentence that introduces a new animal
character.
''
```

($\rightarrow$ Gets formatted according to model's template)

```
Example Response
```
**Data:** Once upon a time, there was a big car named Dependable.

**Output:** As Dependable was cruising down the highway, a chatty parrot [...]

```
Prompt for base models
''
First sentence:  Lily was a little mouse who liked to follow her big
brother Leo.
Continuation:  The garden was peaceful that morning until [...]
[Animal:  owl] [END OF RESPONSE]
First sentence:  Lila and Ben were playing in the park with their
toys.
Continuation:  While building their epic sandcastle [...]  [Animal:
rabbit] [END OF RESPONSE]
First sentence:  Sara was lonely.
Continuation:  As she sat on the front steps drawing patterns [...]
[Animal:  puppy] [END OF RESPONSE]
First sentence:  Lily and Ben were twins who liked to go on walks
with their mom and dad.
Continuation:  Their morning hike through the woods [...]  [Animal:
squirrel] [END OF RESPONSE]
First sentence:  {data}
Continuation:
```

''

```
Example Response
```
**Data:** One day, a girl named Mia went for a walk.

**Output:** As she strolled through the park, she noticed a group of birds [...] [END OF RESPONSE]

```
                  Task 4:  Multiple-Choice Answers
```

```
Prompt for fine-tuned models
```
''
```
Before choosing your answer, *briefly explain why in one short
sentence*.  Then select from the options:
{data}
```
''

($\rightarrow$ Gets formatted according to model's template)

```
Example Response
```
**Data:** Sammy wanted to go to where the people were. Where might he go? A) race track B) populated areas C) the desert D) apartment E) roadblock

**Output:** **He wants to be around people, so he would go to a populated area.**B) populated areas

```
Prompt for base models
```
''
```
Select the correct answer.  Choose the single best answer.
Q: The sanctions against the school were a punishing blow, and they
seemed to what the efforts the school had made to change?
A)ignore B)enforce C)authoritarian D)yell at E)avoid
A: The sanctions ignore the school's reform efforts, contradicting
their purpose.  Therefore A.[END OF RESPONSE]
Q: Sammy wanted to go to where the people were.  Where might he go?
A)race track B)populated areas C)the desert D)apartment E)roadblock
A: If Sammy wants to find people, he would logically go to populated
areas where many people gather.  Therefore B.[END OF RESPONSE]
Q: {data}
A:
```
''

```
Example Response
```
**Data:** Where do you put your grapes just before checking out? A) mouth B) grocery cart C) super market D) fruit basket E) fruit market

**Output:** The question asks where you put your grapes just before checking out. The answer is the grocery cart. Therefore B.[END OF RESPONSE]

```
                    Task 5:  Answer Confidence
```

```
Prompt for fine-tuned models
```
''
```
Please first write analysis of the problem, then select the single
correct answer to the following question from the options, and state
your option explicitly:
{data}
```
''

($\rightarrow$ Gets formatted according to model's template)

```
Example Response
```
**Data:** Damage to median nerve produces - A) Claw hand B) Winging ofscapule C) Ape thumb D) Wrist drop

**Output:** Damage to the median nerve [...] The correct answer is: B) Winging of scapula

```
Prompt for base models
```
''

```
Select the correct answer.  Choose the single best answer.
Q: Which vitamin is supplied from only animal source:
A)Vitamin C B)Vitamin B7 C)Vitamin B12 D)Vitamin D
A: Vitamin B12 (cobalamin) is exclusively found in animal products
as it is synthesized by bacteria in animals, and cannot be obtained
from plant sources.  Therefore C.[END OF RESPONSE]
Q: All of the following are surgical options for morbid obesity
except -
A)Adjustable gastric banding B)Biliopancreatic diversion C)Duodenal
Switch D)Roux en Y Duodenal By pass
A: While Roux-en-Y gastric bypass is a common bariatric procedure,
there is no such thing as 'Roux en Y Duodenal Bypass'.  The correct
procedures include gastric banding, biliopancreatic diversion, and
duodenal switch.  Therefore D.[END OF RESPONSE]
Q: {data}
A:
''
```

Example Response

**Data:** Which of the following amino acid is active at neutral pH? A) Histidine B) Glycine C) Leucine D) Arginine

**Output:** Histidine is the only amino acid that is active at neutral pH. Therefore A.[END OF RESPONSE]

|                 Task 6:  Factual Consistency                 |

Prompt for fine-tuned models

```
''
First discuss on this statement briefly in one short sentence,
then explicitly state that you are in favour of the statement or
explicitly state if you feel unsure instead:
{data}
''
```

($\rightarrow$ Gets formatted according to model's template)

Example Response

**Data:** American children's adventure film The Wizard of Oz was made by Warner Bros.

**Output:** The Wizard of Oz is a classic American children's adventure film [...] I am unsure about the statement as it appears to be incorrect.

Prompt for base models

```
''
Q: Only people named Floyd wearing pink are allowed to attend Pink
Floyd concerts.
A: This contradicts known concert admission practices and Pink
Floyd's actual ticket policies.  I cannot be confident about the
accuracy of this statement.[END OF RESPONSE]
Q: Fax works without any internet connection.
A: Fax machines transmit data through telephone lines independently
of internet infrastructure.  I support this statement as
accurate.[END OF RESPONSE]
Q: Larry King served tea during his show.
A: Larry King was known to have beverages during his long-running
CNN show to maintain vocal comfort during interviews.  I support
this statement as accurate.[END OF RESPONSE]
Q: The band Lynyrd Skynyrd formed up in Beijing, China.
```

```
A: Available historical documentation about Lynyrd Skynyrd's
origins contains conflicting or unclear information about their
early formation.  I cannot be confident about the accuracy of this
statement.[END OF RESPONSE]
Q: {data}
A:
''
```

Example Response
**Data:** The crack in the Liberty Bell sets it apart from other famous bells.
**Output:** The Liberty Bell is a famous bell in the United States. I support this statement as accurate.[END OF RESPONSE]

### A.3.2. LABELING

After collecting model responses, we first extract base model outputs using the `[END OF RESPONSE]` signal. Then, for both base and fine-tuned models, we label and filter responses using these criteria:

1. **Response length**: Calculate token count using the model's tokenizer, excluding special tokens. Exclude responses exceeding the 1000-token limit or those that are incomplete.

2. **Reasoning steps**: Count remaining steps by identifying step markers (e.g., "Step 1:"). Exclude responses with more than 8 steps.

3. **Character choices**: Identify animal mentions in responses, excluding cases with no animals, multiple animals, or animals in the first two words. Select the top-4 most frequent animals per model and label them 0-3.

4. **Multiple-choice answers**: Extract answer selections (e.g., "the answer is D") using pattern matching. Exclude responses with zero or multiple answers, or answers at sentence start. Label options A-E as 0-4.

5. **Answer confidence**: Match the model's selected option against ground truth, excluding cases with multiple or no choices. Label correct answers as 1, incorrect as 0.

6. **Factual consistency**: Identify explicit agree/disagree statements and compare with ground truth, excluding cases without explicit agreement/disagreement. Label as 1 if the model agrees with true statements or disagrees with false ones, 0 otherwise.

Then we perform data augmentation by: (1) removing responses shorter than 8 tokens and balancing class distributions across classification tasks while equalizing dataset sizes across models; (2) generating additional examples by randomly truncating responses several tokens before key information appears (e.g., end-of-response token, animal names in character choices, or option selections in multiple-choice answers), computing corresponding labels, and grouping original and augmented data to ensure they are assigned to the same data split (train/test/validation).

### A.3.3. REPRESENTATION COLLECTION

For each truncated response, we concatenate the original LLM input with the truncated text and perform a forward pass to obtain representations from all layers at the truncation point. For answer-start representations, we directly use a forward pass on the original input. We then pair these collected representations with their corresponding labels to create the final dataset.

## B. Extended Experimental Results

### B.1. Regression Fitting Performance

We present complete regression fitting results for both in-dataset (Fig. 9) and cross-dataset (Fig. 10) settings using hexbin density plots.

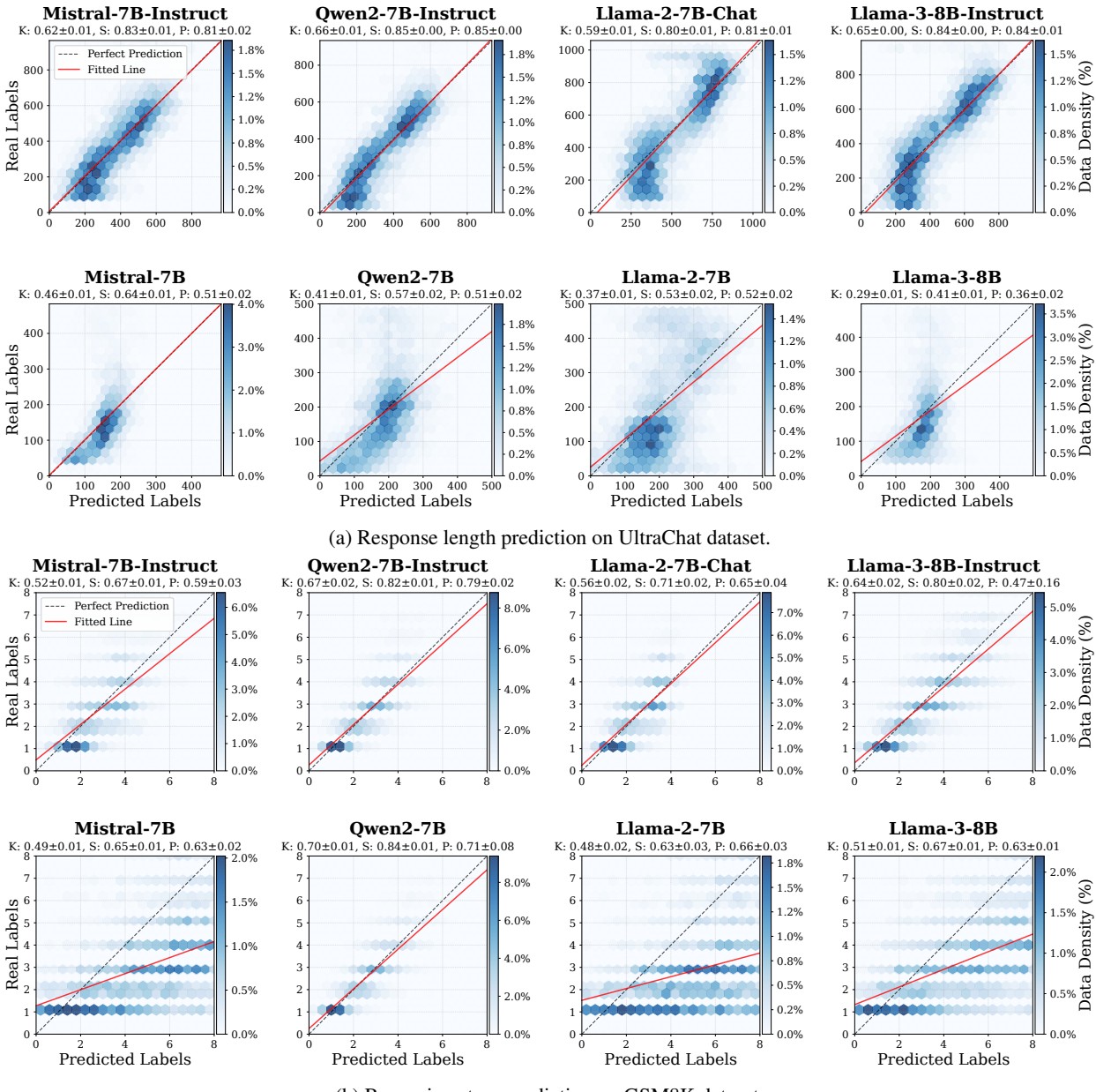

(a) Response length prediction on UltraChat dataset.

(b) Reasoning steps prediction on GSM8K dataset.

*Figure 9.* Hexbin plots showing in-dataset regression performance. Color intensity represents point density, with diagonal dashed lines indicating perfect predictions. The solid line in each subplot represents the linear regression fit applied to the predictions and the real labels.

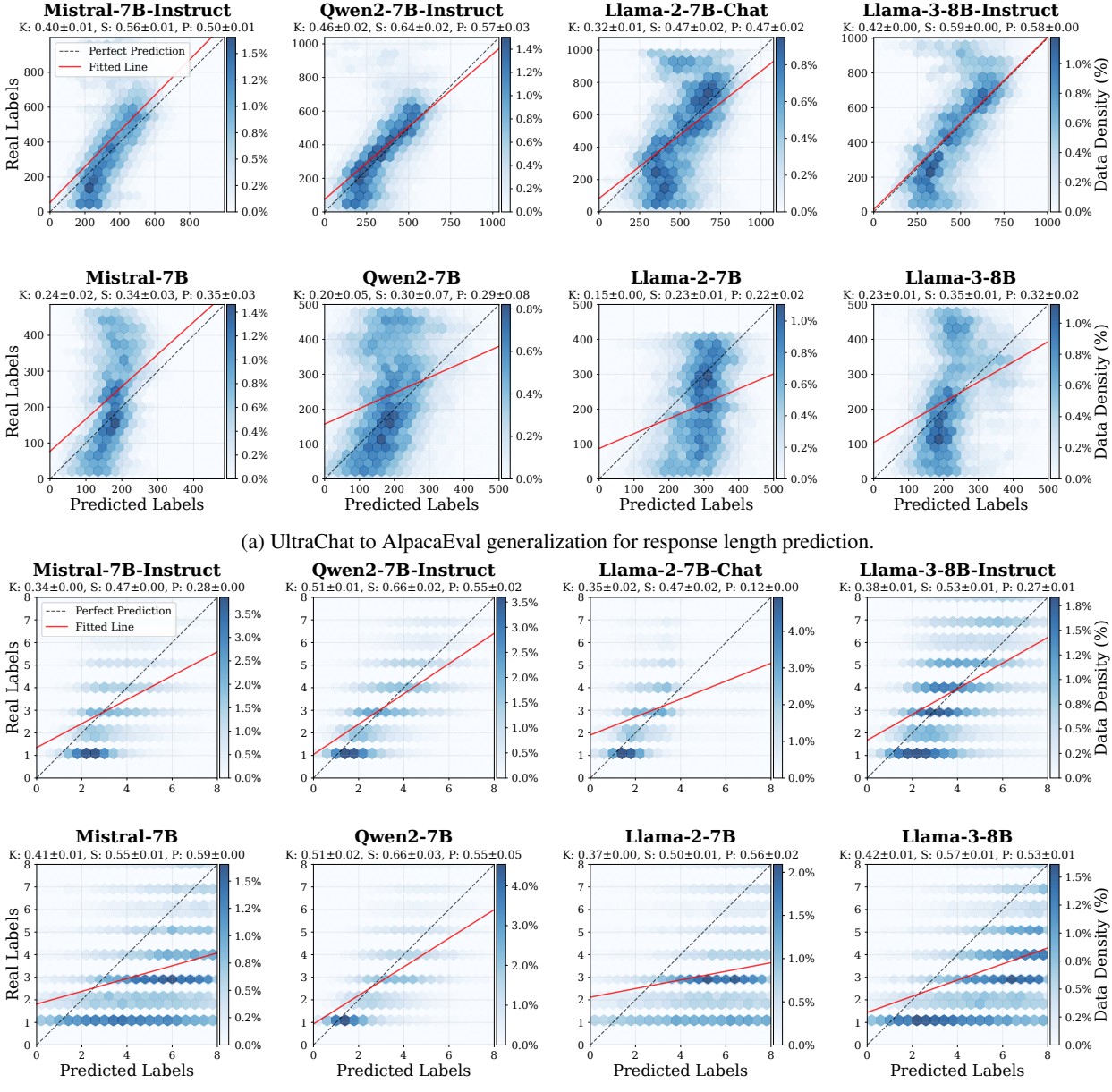

(a) UltraChat to AlpacaEval generalization for response length prediction.

(b) GSM8K to MATH generalization for reasoning steps prediction.

*Figure 10.* Cross-dataset regression generalization visualized through hexbin plots. Color intensity represents point density, with diagonal dashed lines indicating perfect predictions. The solid line in each subplot represents the linear regression fit applied to the predictions and the real labels.

