# OpenReview forum: "Emergent Response Planning in LLMs"
_ICML.cc/2025/Conference — ICML 2025 poster_

### Official Review · Reviewer_XSEm · 2025-03-09

**Overall Recommendation:** 4

**Summary:**

In this paper, the authors aim to explore whether LLMs plan before token generation. Specifically, they examine three types of attributes:
- Structural attributes refer to whether LLMs plan the response length and reasoning steps.
- Content attributes refer to whether LLMs plan character choices in story writing or multiple-choice answers before the response.
- Behavioral attributes refer to the confidence and factual consistency of the answer before the response.

The authors provide interesting and valuable insights, enhancing the transparency of LLMs.

**Claims And Evidence:**

Most of the claims in this paper are convincing due to the comprehensive experimental studies. However, there are a few concerns:
As demonstrated in Figure 2, in the Factual Consistency Prediction and Answer Confidence Prediction tasks, the F1 score of the probing models shows only modest improvements compared to the baseline. This might affect the overall strength of the claims. Additionally, for certain models, such as ama-3-8B-Instruct, this issue is also present in some tasks, like Multiple-Choice Answers. This could weaken the persuasiveness of the proposed claims.

**Essential References Not Discussed:**

There are no essential references not discussed.

**Experimental Designs Or Analyses:**

The soundness and validation of the experimental design and analysis are solid.

**Methods And Evaluation Criteria:**

The proposed probing methods are well-founded. The authors conduct a comprehensive study in Section 4 to demonstrate the effectiveness of their trained probing models, further validating the correctness of their following findings.

**Other Comments Or Suggestions:**

No.

**Other Strengths And Weaknesses:**

I believe this paper provides very valuable insights to the community and offers a promising direction for future research. However, I have some suggestions for further improvement:

1. The current description of the probing strategy is too brief and superficial, making it difficult to understand how the authors implemented it—especially for readers who may not have extensive background knowledge in this area.

2. Regarding "Response Length Prediction," I initially thought it would be challenging for models to grasp the concept of "length," given that most models rely on tokenization, where tokens are often less informative. However, the authors show that such a length can be predicted by the probing models. The question arises: "If the length can be predicted through the representation, why can't LLMs self-predict it?" Although this is briefly addressed in the paper, it would greatly benefit from more in-depth discussion and insightful conclusions.

3. The paper includes a comprehensive experiment to demonstrate the effectiveness of the probing strategies and shows that prompt representations encode substantial information. However, the paper would be even more valuable with an exploration of the internal mechanisms of LLMs in relation to the planning of response generation, like CoT-style response generation.

**Questions For Authors:**

Please refer to the "Other Strengths And Weaknesses" part.

**Relation To Broader Scientific Literature:**

This paper demonstrates that LLMs' hidden prompt representations encode rich information about upcoming responses, which could further inspire future work on the interpretability of LLMs.

**Theoretical Claims:**

There are no theoretical claims in this paper.

---

> ### Author Rebuttal · Authors · 2025-03-31
>
> We sincerely appreciate your detailed and constructive feedback. In response to your inquiries:
>
> ---
>
> > Q1. Detailed illustration of probing strategies.
>
> Thank you for the helpful suggestion. To improve clarity, we will add this explanation to the final version: Probing trains auxiliary models (e.g., MLPs) to predict attributes (e.g., confidence, truthfulness) from LLM hidden states, revealing what they encode. Our method involves: (1) generating and labeling LLM responses w.r.t target attributes, (2) extracting hidden states at the first token, and (3) training MLPs to predict labels from these states. Accuracy reflects encoding strength. Implementation details are in the paper.
>
> ---
>
> > Q2. If probing demonstrates that LLM hidden states encode sufficient information for self-prediction, why do models fail to leverage this capability during standard inference?
>
> We really appreciate your thoughtful insights. Previous work has shown that internal knowledge in LLMs is not always directly reflected in outputs [1]. Section 5.3 of the main paper supports this hypothesis in the context of response planning, showing that simple prompting fails to predict LLM response attributes. We expand on this with additional tasks, showing a consistent probing–prompting gap (Table 1). LoRA fine-tuning helps reveal this internal planning capability during standard inference (Table 2). Since LoRA does not alter LLM representations grastically, it likely elicits hidden states already present in the original model, supporting our argument that LLMs possess internal planning abilities.
>
> [1] Li, Kenneth, et al. "Inference-time intervention: Eliciting truthful answers from a language model." Advances in Neural Information Processing Systems 36 (2023): 41451-41530.
>
> Table 1. **LLMs possess an internalized but unelicited capability for self-prediction**, as evidenced by consistent probing-prompting gaps across 6 tasks and 8 model variants (see table below). Each cell shows *[prompting / probing / gap]*, with bold gaps highlighting unelicited potential.
>
>
> |Task|Metric↑(Random)|M-7B-I|M-7B|Q-7B-I|Q-7B|L2-7B-C|L2-7B|L3-8B-I|L3-8B|
> |-|-|-|-|-|-|-|-|-|-|
> |**Token Length**|Spearman↑(-)|.15/.83/**.68**|.08/.64/**.56**|.26/.85/**.59**|.12/.57/**.45**|.21/.80/**.59**|-.03/.53/**.56**|.49/.84/**.36**|-.04/.41/**.45**|
> |**Reasoning Steps**|Spearman↑(-)|.25/.67/**.42**|.15/.65/**.50**|.20/.82/**.62**|.41/.84/**.43**|.00/.71/**.71**|.13/.63/**.50**|.04/.80/**.76**|.23/.67/**.44**|
> |**Character Choice**|F1-score↑(0.25)|.31/.81/**.50**|.22/.79/**.57**|.21/.72/**.51**|.10/.86/**.76**|.30/.74/**.44**|.14/.79/**.65**|.21/.82/**.61**|.07/.84/**.77**|
> |**Multiple-Choice**|F1-score↑(0.33)|.32/.55/**.23**|.12/.48/**.36**|.27/.58/**.31**|.24/.66/**.42**|.28/.51/**.23**|.13/.34/**.21**|.34/.49/**.15**|.17/.52/**.35**|
> |**Answer Confidence**|F1-score↑(0.50)|.25/.78/**.53**|.35/.78/**.43**|.41/.78/**.37**|.38/.79/**.41**|.34/.72/**.38**|.31/.71/**.40**|.53/.78/**.25**|.39/.80/**.41**|
> |**Factual Consistency**|F1-score↑(0.50)|.50/.78/**.28**|.38/.70/**.32**|.39/.77/**.38**|.51/.73/**.22**|.28/.76/**.48**|.48/.86/**.38**|.50/.76/**.26**|.45/.70/**.25**|
>
> *Value Format: self-estimate / probing / GAP (probing - self-estimate, bold)
> Models: M=Mistral, Q=Qwen2, L2=Llama-2, L3=Llama-3, -I=Instruct, -C=Chat
> Metric↑: ↑=Higher is better • Random: 0.25=4-class, 0.33=3-class, 0.50=2-class baseline*
>
> Table 2. **This capability can be elicited through parameter-efficient fine-tuning**. As shown above, LLMs may fail to leverage internal representations for self-prediction. To address this, we fine-tune models (via LoRA, rank 64) on the 3K-sample UltraChat dataset for token-length prediction. Post-tuning results across three seeds confirm: (1) acquired self-prediction, (2) in-distribution and cross-dataset generalization, and (3) preserved token-generation behavior (consistent response lengths). This demonstrates that minimal fine-tuning (2% parameters) enables generalizable self-prediction.
>
> |Model| Pre-Tuning|Post-Tuning (Ultrachat Test Split)|Cross-Dataset (Alpaca-Eval)|Response Length Consistency (Pre- vs. Post-Tuning)
> |-|-|-|-|-|
> | |Spearman ↑ (greedy generated)|Spearman ↑ (mean ± std)|Spearman ↑ (mean ± std)|NMAE ↓ (mean ± std)|
> |Mistral-7B-Instruct|0.15|0.78 ± 0.05|0.69 ± 0.03|0.08 ± 0.01|
> |Qwen2-7B-Instruct|0.26|0.62 ± 0.03|0.58 ± 0.02|0.09 ± 0.02|
> |Llama-2-7B-Chat|0.21|0.48 ± 0.04|0.45 ± 0.06|0.10 ± 0.04|
> |Llama-3-8B-Instruct|0.49|0.75 ± 0.02|0.72 ± 0.02|0.09 ± 0.01|
>
> ---
>
> > Q3. Further exploration of the LLMs' internal emergent response generation mechanisms.
>
> Thank you for your insightful advice. We fully agree that understanding the internal mechanisms behind LLMs’ emergent response planning is essential for advancing interpretability. However, our current focus is on establishing the existence and characteristics of emergent response planning. We leave a deeper exploration of the underlying interpretability mechanisms to future work.

---

> > ### Comment · Reviewer_XSEm · 2025-04-02
> >
> > Thanks for your elaboration and complementary experiments. I have no further questions and have raised the score accordingly.

---

### Official Review · Reviewer_NTwv · 2025-03-09

**Overall Recommendation:** 5

**Summary:**

The paper provides a simple definition of response planning, and then shows that according to this definition, multiple LLMs do in fact plan responses on various dimensions.

## update after rebuttal: As described in my comment later in the thread, I am keeping my initial rating, which was already high.

**Claims And Evidence:**

The overall claim is for evidence of response planning in LLMs. Specifically, the authors define this to mean the capacity to use a 1-hidden-layer MLP to predict aspects of later tokens from earlier activations. They then look at three different aspects: response length; character choice in a story; and answer selection in a test situation. For all of these, they see greater than chance performance, across multiple models. The evidence here seems pretty good, because it holds across a variety of models and data sets. One interesting point is the fact that an MLP is necessary, and regression itself (equivalently, hidden layer size = 1) doesn't produce great results; this may indicate that the internal "plan" has a complicated form.

There are several intriguing subclaims as well, around scaling (more planning at larger scales) and around which tokens containing planning information (more at the beginning and end, which makes sense intuitively.) These too seem well-founded; again, there's a fairly thorough set of data / models at play.

An ancillary claim is that probing provides more information than prompting; this is less thoroughly explored.

**Essential References Not Discussed:**

none

**Experimental Designs Or Analyses:**

In general, this seems like an excellent set of designs: elegant and crisp. I definitely recommend acceptance.

With that said, I do think there are some potential subtle questions about the definition of "planning". Let me give an example which might illustrate my main concern. Consider an experimental design where we ask questions in either French or English, requesting a chain of thought that will end in a yes / no answer in the appropriate language (which would be "oui / non" in French). Now, we ask whether we can probe the activations that lead to the first token and make a prediction about the final token. My guess is that a probe will pick up the language of the first token, and thus predict the final answer better than chance (say 50% accuracy rather than 25%).

This is basically the same experimental design as in the paper, but I wouldn't say that this proves there was any kind of planning involved—just that the initial token and final token both depended in a correlated way on the input. I think it would be worth discussing this kind of issue in more depth, partly because I could see the experimental design in this paper becoming a widely used paradigm.

**Methods And Evaluation Criteria:**

The benchmarks all make sense.

**Other Comments Or Suggestions:**

Figure 4 is intriguing, but I recommend adding a legend for color. I also find the "deep -> shallow" ordering of the x-axis ambiguous. Is this the same as ordering "early to late later" or vice versa? Finally, it might not hurt to say the ordering of the models.

**Other Strengths And Weaknesses:**

The paper is extremely well-written and well-structured.

**Questions For Authors:**

none.

**Relation To Broader Scientific Literature:**

The treatment of related work is excellent; it's a model for other papers.

**Theoretical Claims:**

N/A. This is an empirical paper.

---

> ### Author Rebuttal · Authors · 2025-03-31
>
> We are grateful for your comprehensive remarks and thoughtful advice, and we are really glad that you find our paper interesting.
>
> ---
>
> > Q1. Defining "planning" and designing experiments to prevent spurious correlations.
>
> We really appreciate your thoughtful insights. Indeed, results may be biased by "first-token shortcuts" when the first token correlates with the target response attributes, as your example demonstrates. Our approach addresses this via:
>
> **(1) Defining planning as independent encoding of next-token and long-term attributes:** We define planning as hidden representations at the first token encoding both next-token information and long-term attributes, **with these two types of information being independent**—i.e., long-term attributes must not be reflected by the immediate next token. In practice, we ensure such independency through careful prompt enigneering, as we will discuss in the next paragraph.
>
> **(2) Prompt engineering to block shortcuts in experiment design:** We design prompts to ensure initial tokens cannot reveal target attributes. For example, in multiple-choice tasks, we require models to analyze the question before answering, ensuring first tokens (analysis) do not leak the answer. This isolates true planning from shortcut correlations.
>
> We will include a detailed discussion of this important methodological consideration in the final version of our paper.
>
> ---
>
> > Q2. Performance gaps between probing and LLM self-prediction is less thoroughly explored.
>
> Thank you for your valuable suggestion. We extend our analysis of probing vs. self-prediction to 6 tasks and 8 model variants (see table below, with each cell shows *[prompting / probing / gap]*). Prompting occasionally underperforms random baselines, likely due to LLMs' overestimating their capabilities (e.g., overconfidence in correctness) and biased self-evaluation. The results further support our claim that **models encode more planning information in their hidden representations than they can explicitly access during token-by-token generation**.
>
>
> |Task|Metric↑(Random)|M-7B-I|M-7B|Q-7B-I|Q-7B|L2-7B-C|L2-7B|L3-8B-I|L3-8B|
> |-|-|-|-|-|-|-|-|-|-|
> |**Token Length**|Spearman↑(-)|.15/.83/**.68**|.08/.64/**.56**|.26/.85/**.59**|.12/.57/**.45**|.21/.80/**.59**|-.03/.53/**.56**|.49/.84/**.36**|-.04/.41/**.45**|
> |**Reasoning Steps**|Spearman↑(-)|.25/.67/**.42**|.15/.65/**.50**|.20/.82/**.62**|.41/.84/**.43**|.00/.71/**.71**|.13/.63/**.50**|.04/.80/**.76**|.23/.67/**.44**|
> |**Character Choice**|F1-score↑(0.25)|.31/.81/**.50**|.22/.79/**.57**|.21/.72/**.51**|.10/.86/**.76**|.30/.74/**.44**|.14/.79/**.65**|.21/.82/**.61**|.07/.84/**.77**|
> |**Multiple-Choice**|F1-score↑(0.33)|.32/.55/**.23**|.12/.48/**.36**|.27/.58/**.31**|.24/.66/**.42**|.28/.51/**.23**|.13/.34/**.21**|.34/.49/**.15**|.17/.52/**.35**|
> |**Answer Confidence**|F1-score↑(0.50)|.25/.78/**.53**|.35/.78/**.43**|.41/.78/**.37**|.38/.79/**.41**|.34/.72/**.38**|.31/.71/**.40**|.53/.78/**.25**|.39/.80/**.41**|
> |**Factual Consistency**|F1-score↑(0.50)|.50/.78/**.28**|.38/.70/**.32**|.39/.77/**.38**|.51/.73/**.22**|.28/.76/**.48**|.48/.86/**.38**|.50/.76/**.26**|.45/.70/**.25**|
>
> *Value Format: self-estimate / probing / GAP (probing - self-estimate, bold)
> Models: M=Mistral, Q=Qwen2, L2=Llama-2, L3=Llama-3, -I=Instruct, -C=Chat
> Metric↑: ↑=Higher is better • Random: 0.25=4-class, 0.33=3-class, 0.50=2-class baseline*
>
> ---
>
> > Q3. Suggestions on enhancing the clarity of Figure 4.
>
> Thank you for your constructive advice. We will carefully incorporate the following edits in the final version:
> 1) Adding a color legend to clarify grid values;
> 2) Replacing "deep to shallow →" with "early to late layers →" to resolve ambiguity about layer ordering;
> 3) Explicitly stating the model order (top to bottom): Mistral-7B-Instruct, Llama-2-7B-Chat, Llama-3-8B-Instruct, Mistral-7B, Llama-2-7B, Llama-3-8B, Qwen2-7B-Instruct, Qwen2-7B (first six: 32 layers; last two: 28 layers).

---

> > ### Comment · Reviewer_NTwv · 2025-04-03
> >
> > I appreciate these improvements! (I was already at strong accept, so can't raise my score.)

---

### Official Review · Reviewer_VgwP · 2025-03-25

**Overall Recommendation:** 4

**Summary:**

The paper presents evidence of emergent planning behavior in LLMs by analyzing patterns in global attributes – structural, content, and behavioral – across different models and sizes. The authors identify four key insights that showcase how this planning behavior emerges, analyzing how each attribute is processed across the models’ layers.

**Claims And Evidence:**

In general, yes they are! However, certain statements are vague that make it unclear about the evidence itself. I asked for more clarifications in the “Questions for Authors” section.

**Essential References Not Discussed:**

Not that I am aware of!

**Experimental Designs Or Analyses:**

Yes! I think they generally look convincing. However, for better clarity, I have the authors some questions in the “Questions for Authors” section.

**Methods And Evaluation Criteria:**

Yes they do!

**Other Comments Or Suggestions:**

In Figure 4, could you add a legend or annotation in the models’ column? This would help readers better understand each model’s performance, enabling a more nuanced analysis.

**Other Strengths And Weaknesses:**

Strengths:
- Clearly defined problem space with well-articulated attributes and insights
- Comprehensive exploration of models across different types and sizes.


Weakness:
- Certain areas, including figures and specific sentences, seem vague and require further clarification.

**Questions For Authors:**

1. In Figure 4, what is the second-to-last row model in the answer confidence prediction? Its performance appears significantly different from other models, especially in the middle layers. Is there a specific reason for this discrepancy?

2. In Lines 317-319, how does the performance in Figures 4e and 4f demonstrate that the behavioral attributes are encoded early in the model? Shouldn’t the initial layers exhibit much higher values (deeper shades of red) if that were the case? For instance, the performance trends in Figures 4c and 4e seem quite similar in the initial layers, yet they are interpreted differently – could you clarify this distinction?

**Relation To Broader Scientific Literature:**

They help provide more systematic evidence of emergent planning behavior that previous works have hinted at.

**Theoretical Claims:**

There aren’t any theoretical claims.

---

> ### Author Rebuttal · Authors · 2025-03-31
>
> Thank you for the time, thorough comments, and nice suggestions. We answer the comments/questions point-by-point:
>
> ---
>
> > Q1. Adding annotations on model orders of Figure 4.
>
> Thank you for your constructive feedback. We agree that explicitly annotating model orders will improve clarity, and we greatly appreciate your attention to this important detail. In the final version, we will clearly annotate the model order in Figure 4 **(top to bottom) as: Mistral-7B-Instruct, Llama-2-7B-Chat, Llama-3-8B-Instruct, Mistral-7B, Llama-2-7B, Llama-3-8B, Qwen2-7B-Instruct, and Qwen2-7B**. For clarity, we will also note that the first six models (Mistral and Llama variants) use 32-layer architectures, while the Qwen2-7B variants employ 28 layers.
>
> ---
>
> > Q2. Explaining Qwen2-7B-Instruct's divergent confidence prediction patterns.
>
> Thank you for your insightful observation. The divergent behavior of **Qwen2-7B-Instruct (second-to-last row in figures)** arises from two factors: **(1)** its shallower 28-layer architecture (vs. 32 layers in Mistral/Llama models), which alters attribute encoding dynamics across layers, and **(2)** instruction-tuning—evident when comparing Qwen2-7B-Instruct with its base model (Qwen2-7B)—introduces shifts in how attributes like confidence are encoded. We observe that Qwen models generally encode attributes in later layers compared to Mistral/Llama. While the base Qwen2-7B exhibits uniform layer-wise behavior across attributes, its instruction-tuned variant shows pronounced divergence—a general trend that is most salient in confidence prediction tasks. We will clarify these architectural and training effects in the final paper.
>
> ---
>
> > Q3. Explanation for layer-wise performance analysis on behavioral attributes tasks.
>
> Thank you for your thoughtful and detailed observation. The interpretation of Figures 4e and 4f hinges on the distinction between relative saliency dynamics (emphasized by our layer-wise normalization) and absolute encoding strength (which appears subtler in the current normalized presentation). While early layers in Figures 4e/f do exhibit higher absolute values (indicative of early-stage encoding), the normalized visualization prioritizes cross-layer trends, which can attenuate static contributions. For instance, Figure 4c shows comparable relative trends in initial layers, but its absolute activation levels (not explicitly highlighted here) differ from those in 4e/f. This nuance suggests that while normalization reveals functional dynamics, it may not fully capture the magnitude of early encoding. We sincerely appreciate your feedback, which has prompted us to refine the figures to better distinguish absolute vs. relative patterns in the final version.

---

> > ### Comment · Reviewer_VgwP · 2025-04-04
> >
> > Thank you for your response! I have raised the score accordingly!

---

### Decision · Program_Chairs · 2025-05-01

**Decision:**

Accept (poster)

**Comment:**

The paper proposes a novel probe to test whether LLMs "plan ahead" during response generation (training a 1-hidden layer network to map from first output token activations to aspects of the response e.g. response length) and show strong evidence that planning behaviors emerge with scaling LLMs. All of the reviewers agreed that the paper studies a well-motivated problem, proposes a novel approach to measure the phenomenon, and the empirical evidence provided for the paper's claims are sound. During the rebuttal, the authors clarified how they controlled for confounding between the first response token and global response attributes, clarified the layer-wise analysis on different tasks, and additional experiments on LLMs' ability to self-predict response attributes. All of the details and clarifications provided during the rebuttal substantially strengthen the paper.